# MukB Is a Gene Necessary for Rapid Proliferation of *Vibrio vulnificus* in the Systemic Circulation but Not at the Local Infection Site in the Mouse Wound Infection Model

**DOI:** 10.3390/microorganisms9050934

**Published:** 2021-04-27

**Authors:** Takashige Kashimoto, Kohei Yamazaki, Takehiro Kado, Kaho Matsuda, Shunji Ueno

**Affiliations:** Laboratory of Veterinary Public Health, School of Veterinary Medicine, Kitasato University, 23-35-1 Higashi, Towada, Aomori 034-8628, Japan; kyamazak@vmas.kitasato-u.ac.jp (K.Y.); dv15003q@st.kitasato-u.ac.jp (T.K.); vm12126f@st.kitasato-u.ac.jp (K.M.); ueno@vmas.kitasato-u.ac.jp (S.U.)

**Keywords:** *Vibrio vulnificus*, sepsis, MukB

## Abstract

*Vibrio vulnificus* causes rapid septicemia in susceptible individuals who have ingested contaminated foods or have open wounds exposed to seawater contaminated with the bacteria. Despite antibiotic therapy and aggressive debridement, mortality from septicemia is high. In this study, we showed that MukB mutation (*mukB*::Tn) affected the proliferation of *V. vulnificus* in the systemic circulation but not at the inoculation site in the wound infection model. A comparison of *mukB*::Tn with WT and a *mukB* complement strain (*mukB*::Tn/p*mukB*) on the bacterial burden in the muscle at the infection site showed that spreading and proliferation of the *mukB*::Tn strain was similar to those of the other strains. However, the bacterial burden of *mukB*::Tn in the spleen was reduced compared to that of the WT strain in the wound infection model. In a competition experiment, we found a lower bacterial burden of *mukB*::Tn in the spleen than that of the WT strain infecting the systemic circulation. Here, we report on a gene required for the rapid proliferation of *V. vulnificus* only in the systemic circulation and potentially required for its survival. Our finding may provide a novel therapeutic target for *V. vulnificus* septicemia.

## 1. Introduction

*Vibrio vulnificus* is a Gram-negative bacterium that can cause severe septicemia [1,2,3]. People become infected when they eat contaminated seafood, have open wounds exposed to contaminated seawater or handle a contaminated marine product [4]. Bullae and necrotizing fasciitis develop in any of these cases. Even after treatment with antibiotics, aggressive surgery, or both, *Vibrio vulnificus* rapidly proliferates in some patients [5]. Thus, it is essential to identify virulence factors necessary for the rapid proliferation of *V. vulnificus* after its invasion into the systemic circulation. These factors could be targeted to develop effective treatments for septicemia. In a murine infection model, several factors have already been reported: a capsular polysaccharide, lipopolysaccharide, repeats-in- toxins, an iron acquisition system, and chemotactic ability [6,7,8,9]. However, the mechanisms causing severe and rapid septicemia resulting from *V. vulnificus* infection have not been fully understood.

The structural maintenance of chromosome (SMC) complex in *Escherichia coli* comprises chromosome partition proteins MukB, MukE, and MukF [10,11]. MukB is the core subunit of the SMC complex and forms homodimers. Each MukB monomer has two ATP binding pockets, which are necessary to bind MukE and MukF complex and cause extrusion of the DNA loop by a conformational change of MukB from an open state to a closed state [12]. One MukB mutant of *E. coli* was identified as a temperature-sensitive strain that cannot grow above 30 °C [13]. It has been hypothesized that a conformational change of the MukB dimer between the open state and closed state is dependent on ATP binding and temperature fluctuations [14]. Most studies have focused on the architecture and stoichiometry of MukB. This study aimed to identify the role of MukB during *V. vulnificus* infection and its effect on the host. Our results showed that MukB is one of the necessary genes for rapid proliferation of *V. vulnificus* in the systemic circulation but not at the inoculation site in the wound infection model.

## 2. Materials and Methods

### 2.1. Ethics Statement

All animal studies were carried out in strict accordance with the Guidelines for Animal Experimentation of the Japanese Association for Laboratory Animal Science (JALAS). The animal experimentation protocol was approved by the president of Kitasato University based on the judgment of the Institutional Animal Care and Use Committee of Kitasato University (Approval No. 15-156).

### 2.2. Bacteria

*V. vulnificus* clinical isolated strain CMCP6 (WT) and nonencapsulated strain E4 (Environmental isolate from seafood) were cultured aerobically in Luria-Bertani (LB) broth or on LB agar at 37 °C. When required, the medium was supplemented with rifampicin (50 µg/mL) to selectively grow for *V. vulnificus*, chloramphenicol (10 µg/mL) to maintain the pACYC, or ampicillin (100 µg/mL) to maintain the pXen-13 plasmid for in vivo imaging systems. The pACYC plasmid was used for *mukB* complementation.

### 2.3. Generation of Transposon Mutants and Identification of the Tn Insertion Gene

The construction of a library containing 63 mutants with a transposon tagged by a unique sequence was performed as previously described [15]. Briefly, *E. coli* BW19795 with signature-tagged mini-Tn5Km2 in pUT were combined with *V. vulnificus* on a nitrocellulose Hybond C membrane (GE Healthcare, Tokyo, Japan) for conjugation, placed onto an M9 agar plate, and incubated at 25 °C. The bacterial suspension was plated onto TCBS agar containing 100 µg/mL kanamycin and incubated overnight at 37 °C for selection. Each signature-tagged transposon insertion mutants of *V. vulnificus* were grown in Luria-Bertani (LB) medium containing 100 μg/mL of kanamycin for 12 h at 37 °C in 96 well plate separately. We checked each bacterial growth by measuring optical density at 600 nm (OD_600_) with a microplate reader (Sunrise/TECAN Japan, Kanagawa, Japan), and the mutants were pooled, washed with LB medium without any anti-biotics, and used as an input pool. Mice were subcutaneously inoculated with 10^6^ colony forming unit (CFU) of the input pool into right caudal thighs. The infected mice were carefully monitored and sacrificed by sevoflurane (Wako pure chemical industries, Osaka, Japan) inhalation approximately 24 h after infection when they displayed critical symptoms that are directly associated with death, such as deep hypothermia, and the output pool was collected from murine spleens. Selection of attenuated mutants by STM was performed in triplicate for each of 86 libraries.

For tag-specific dot hybridization, 10 μM of the target DNA, comprising the signature-tagged sequence region of each transposon, was blotted onto Hybond-N+ membrane (GE Healthcare) and fixed with CL-1000 Ultraviolet Crosslinkers (UVP, Upland, CA, USA). DNA probes were amplified by polymerase chain reaction (PCR) with Dig-labeled primers: Dig 1 (5′-Dig-CAT GGT ACC CAT TCT AAC-3′) and Dig 2 (5′-Dig-TAC CTA CAA CCT CAA GCT-3′). PCR was performed in a 50-μL reaction mix containing 25 μL of 2 × PCR Buffer for KOD FX Neo, 5 μL of 2 mM dNTPs, 2 μL of primer mix (0.3 μM final concentration of each primer), 0.5 μL KOD FX Neo polymerase (0.5 unit, Toyobo, Osaka, Japan), 1 μL DNA template (about 200 ng genomic DNA), and distilled water. Thermal cycling conditions were as follows: (i) 5 min at 94 °C; and (ii) 25 cycles of 15 s at 94 °C, 45 s at 59 °C, and 10 s at 68 °C. The hybridization processes were performed in a hybridization oven (MHS-200e/eyelaco, Tokyo, Japan). The transposon inserted sequence was determined via arbitrarily primed PCR using specific primers targeting the transposon: Rev consensus2 I-out 1st (5′-CCA TGG GTA AGA TTG GTT CGA A-3′), consensus1 I-out 1st (5′-GGT ACC TAC AAC CTC AAG CT-3′), and Rev consensus1 I-out for 2nd (5′-AGC TTG GTT AGA ATG GGT ACC-3′), and random primers targeting the *V. vulnificus* genome: Arb1 (5′-GGC CAC GCG TCG ACT AGT CAN NNN NNN NNN GAT AT-3′), Arb3 (5′-GGC CAC GCG TCG ACT AGT CAN NNN NNN NNN TTC AA-3′), Arb4 (5′-GGC CAC GCG TCG ACT AGT CAN NNN NNN NNN CCA CG-3′), Arb5 (5′-GGC CAC GCG TCG ACT AGT CAN NNN NNN NNN ACT GA-3′), Arb6 (5′-GGC CAC GCG TCG ACT AGT CAN NNN NNN NNN ACG CG-3′), Arb7 (5′-GGC CAC GCG TCG ACT AGT CAN NNN NNN NNN TGG CA-3′), and Arb for 2nd (5′-GGC CAC GCG TCG ACT AGT CA-3′). Thermal cycling conditions of first round were as follows: (i) 5 min at 95 °C; (ii) 6 cycles of 20 s at 95 °C, 20 s at 30 °C, and 2 min at 72 °C; (iii) 30 cycles of 20 s at 95 °C, 20 s at 45 °C, and 2 min at 72 °C; and (iv) 5 min at 72 °C. PCR was performed in a 50-μL reaction mix containing 5 μL of 10 × PCR Buffer for Paq5000, 2 μL of 2 mM dNTPs, 2 μL of primer mix (0.5 μM final concentration of each primer), 0.5 μL Paq5000 polymerase (0.5 unit, Stratagene, CA, USA), 1 μL DNA template (about 200 ng genomic DNA), and distilled water. Thermal cycling conditions of second round were as follows: (i) 30 sec at 95 °C; (ii) 30 cycles of 20 s at 95 °C, 30 s at 55 °C, and 4 min at 68 °C; and (iii) 5 min at 68 °C. PCR was performed in a 50 μL reaction mix containing 5 μL of 10 × PCR Buffer for Taq DNA polymerase with ThermoPol, 5 μL of 2 mM dNTPs, 2 μL of primer mix (0.5 μM final concentration of each primer), 0.5 μL Taq DNA polymerase with ThermoPol (0.5 unit, New England Biolabs, MA, USA), 1 μL the PCR product of the first round, and distilled water. The DNA sequences of the PCR products were determined by using the Fasmac (Atsugi, Japan) sequencing service and used to search for sequence homologies in the Kyoto Encyclopedia of Genes and Genomes database (KEGG: www.genome.ad.jp/keg, accessed on 26 March 2021).

### 2.4. Complementation of MukB

The full length of *mukB* was amplified by PCR with the primers pACYC *Bam*HI *mukB* Fw (5′- AGG ATA AAT GGC TAG ATG ATT GAA AGA GGT AAA TAT C -3′) and pACYC *Xho*I *mukB* Rev (5′- CGG GCC CCC CCT CGA TTA TCG CTA TTG AGT TTA -3′) from *V. vulnificus* CMCP6 genome as the template. The amplified DNA was ligated to *Bam*HI and *Xho*I site of pACYC and the sequence was confirmed by DNA sequencing. The *mukB*::Tn mutant was complemented with this full-length *mukB* gene carried by pACYC. The Wt and *mukB*::Tn were also transformed with empty pACYC.

### 2.5. Mice

Five-week-old female C57BL/6 and BALB/c mice were purchased from Charles River Laboratories Japan (Atsugi, Japan). Except for the IVIS experiments, C57BL/6 mice were used for all other experiments in our study. Both C57BL/6 and BALB/c mice were bred and maintained under specific pathogen-free conditions at Kitasato University. The mice were housed in plastic cages in a group and were maintained on a standard laboratory diet (rat chow MF, Oriental Yeast Co., Ltd. Tokyo, Japan) and tap water under a 12 h light and dark cycle. Ambient temperature during the study was maintained at about 21 °C.

### 2.6. In Vitro Growth Curve Analysis

*V. vulnificus* was grown overnight in LB broth containing rifampicin (50 µg/mL) and chloramphenicol (10 µg/mL) with shaking at 37 °C. Bacteria were washed once to remove the antibiotics with PBS (pH 7.2) containing 0.1% gelatin. After washing, the bacteria were diluted with fresh LB broth containing chloramphenicol (10 µg/mL). Routinely, the starting OD_600_ of the cultures were adjusted around 0.01. The cultures were grown with shaking at 163 rpm at 37 °C for 20 h. During this cultivation, aliquots of culture were taken every 5 h, and measured at OD_600_.

### 2.7. Survival Curve Analysis

Overnight cultures (100 μL) were placed in 2 mL of fresh LB broth containing chloramphenicol (10 µg/mL) and incubated for 2 h. After incubation, the OD_600_ of the cultures was adjusted to 1.0. Bacteria were harvested, washed with PBS (pH 7.2) containing 0.1% gelatin, and resuspended in fresh LB broth. Then, 10^6^ CFU/mouse were subcutaneously inoculated into mice (5 weeks, C57BL/6, female, Charles River Laboratories Japan). Data were analyzed for significant differences using the Log-rank (Mantel-Cox) test.

### 2.8. Bacterial Counts in Spleen and Muscles

Overnight cultures (100 μL) were washed once with PBS (pH 7.2) containing 0.1% gelatin, then the bacteria were inoculated into 2 mL of fresh LB medium containing chloramphenicol (10 µg/mL) and incubated for 2 h. Bacteria were harvested, washed with PBS (pH 7.2) containing 0.1% gelatine, and resuspended in fresh LB medium. Then, 10^6^ CFU/mouse were subcutaneously (s.c.) inoculated in mice. Infected mice were sacrificed at defined time points. The collected muscles beneath the inoculation site or spleen were suspended in cold PBS containing 0.1% gelatine, homogenized for 5 s with a lab mixer IKA EUROSTAR digital (IKA, Werke, Germany; 1300 rpm), and centrifuged at 42× *g* for 5 min. The supernatants were plated at 10-fold serial dilutions in duplicate on LB agar containing 50 µg/mL rifampicin and incubated for 12 h at 37 °C. *V. vulnificus* colonies were counted, and bacterial burden was determined by calculating the number of CFU/g.

### 2.9. In Vivo Growth Competition Assay

WT, *mukB*::Tn and *mukB*::Tn/p*mukB* were grown overnight with 50 µg/mL of rifampicin and 10 µg/mL of chloramphenicol containing LB broth with shaking at 37 °C, and for the *mukB*::Tn and *mukB*::Tn/p*mukB*, additionally added the 50 µg/mL of kanamycin. The overnight cultures were diluted with fresh LB broth, which are containing the same antibiotics above, and then cultivate for 2 h. After 2 h cultivation, OD_600_ of the cultures were adjusted 1.0 and diluted ten times. Each mixed inoculum containing equal volume of WT and *mukB*::Tn, and WT and *mukB*::Tn/p*mukB* was inoculated via intravenously to the mice. After 6 h, mice were euthanized by sevoflurane (Wako Pure Chemical Industries). Spleen of these mice were collected and homogenized, then the homogenates were centrifuged (42× *g*, 5 min), and supernatants were plated onto LB agar plates containing 50 μg/mL of rifampicin, and LB agar plates containing 50 μg/mL of rifampicin and kanamycin. After incubation at 37 °C for 24 h, competitive indexes were calculated ((CFU of mutant / CFU of WT recovered from spleen)/(CFU of mutant / CFU of WT present in initial mixture)).

### 2.10. In Vivo Bioluminescent Imaging

The plasmid pXen-13, which contains a bacterial luminescent gene cluster (*luxCDABE*), was transformed into *V. vulnificus* via electroporation. Electroporation was performed in a cuvette with a 0.2 cm electrode gap (Bio-Rad Laboratories, CA, USA). Stable transformants were selected on LB agar containing 100 µg/mL ampicillin. *V. vulnificus* was grown in LB medium supplemented with 100 µg/mL ampicillin with agitation (163 rpm) at 37 °C. Overnight cultures (100 μL) were inoculated into 2 mL of fresh LB medium supplemented with 100 µg/mL ampicillin and incubated for 2 h. Bacteria were harvested, washed with PBS (pH 7.2) containing 0.1% gelatin, and resuspended in fresh LB medium. Then, 10^6^ CFU/mouse were s.c. inoculated in BALB/c mice. Luminescence signals emanating from *V. vulnificus* were imaged at defined time points using an IVIS 200 imaging system (Xenogen/PerkinElmer, MA, USA) with a 1 min exposure time. The total photons emitted were acquired using the Living Image software package. Mice were anesthetized in chambers containing 2.0% isoflurane inhalant (Pfizer, Tokyo, Japan).

### 2.11. Statistical Analysis

Statistical analysis was performed using GraphPad Prism (GraphPad Software, CA, USA). Survival curves were analyzed using the log-rank test. For analyzing the bacterial burdens in the spleen, Dunn’s multiple comparisons test was used. Statistical differences between the groups in competition assay were analyzed using the Mann-Whitney *U* test. A P value less than 0.05 was considered significant, and Significance values are indicated as follow: *, *p* < 0.05.

## 3. Results

### 3.1. The MukB Mutation Delayed Progression to Death in the Wound Infection

To understand the rapid proliferation mechanisms of *V. vulnificus* in vivo, we utilized signature-tagged mutagenesis (STM) [8] in a mouse wound infection model. Using STM, we identified two mutant clones of MukB that contained a transposon at different positions into the MukB open reading frame. The structure-function relationship of *E. coli* MukB has been studied extensively [10,16,17]. *E.coli* MukB plays a central role in chromosome condensation and segregation by using the energy of ATP hydrolysis when MukB binds with MukE and MukF. By using a local alignment search tool, we found the amino acids sequences aligned between MukB in *E. coli* K12 (1486 a.a.) and MukB in *V. vulnificus* CMCP6 (1487 a.a.). There was 75% similarity in 98% of coverage in both proteins, and that the Walker A motif, C motif, Walker B motif, and D loop, which forms the two ATP binding pockets in the head domain of MukB, were completely conserved (data not shown). Each *mukB*::Tn mutant, which was obtained by our STM assay, resulted in truncated protein from residues1-918 lysine (*mukB*::Tn #1) and 1-1269th glycine (*mukB*::Tn #2), respectively (Figure 1A). Both clones were expressed the Walker A motif and hinge domain of MukB but were not expressed in the C motif, Walker B motif, and D loop (Figure 1A), predicting that in both clones, the change in conformation from the open state to the closed state was blocked (Figure 1B). The MukBEF complex is not expected to form in both *mukB*::Tn clones because the C-terminal domain was not expressed (Figure 1B). We used the *mukB*::Tn #1 as the *mukB*::Tn throughout our study. Firstly, we compared in vitro growth of *mukB*::Tn with those of the parent strain of *V. vulnificus* CMCP6 (WT) and the *mukB* complement strain (*mukB*::Tn/p*mukB*). The tested strains of *V. vulnificus* showed consistently similar growth patterns at 37 °C in Luria-Bertani broth by measuring optical density at 600 nm (OD_600_) (Figure 2A). Therefore, we compared the survival time of mice subcutaneously inoculated with 10^6^ colony forming units (CFU) into the right caudal thigh of one of the following strains: WT, *mukB*::Tn, or *mukB*::Tn/p*mukB*. The median survival time of mice inoculated with the strains was 13.7 h for WT, 14.3 h for *mukB*::Tn/p*mukB*, and 18.3 h for *mukB*::Tn. This increase in the survival time in mice infected with *mukB*::Tn compared to WT and *mukB*::Tn/p*mukB* strains (Figure 2B) suggests that MukB is one of the necessary genes for the rapid proliferation of *V. vulnificus* in vivo.

### 3.2. MukB Is the Necessary Gene for Proliferation of V. vulnificus in the Systemic Circulation

We investigated whether the MukB mutation affected the rapid proliferation of *V. vulnificus* in vivo. Mice were subcutaneously inoculated with WT, *mukB*::Tn, or *mukB*::Tn/p*mukB*. At 12-h post-inoculation, bacterial burdens in the spleen were analyzed. The *mukB*::Tn had remarkably lower CFU counts than those of WT and *mukB*::Tn/p*mukB* (Figure 3A). The mean CFU in *mukB*::Tn was about 1528 times lower that of WT. This result suggests that MukB is the necessary gene for proliferation of *V. vulnificus* in systemic circulation. To confirm this data, we performed a competitive assay between WT and *mukB*::Tn in the systemic circulation. A mixed inoculum containing equal CFU counts of WT and *mukB*::Tn or WT and *mukB*::Tn/p*mukB* was injected intravenously into mice. After 6 h of post-inoculation, the competitive indices were calculated as described in Materials and Methods. The ratio of *mukB*::Tn to WT was 0.033 ± 0.09 (Mean ± Standard error of means; SEM), while the competitive index of WT vs. *mukB*::Tn/p*mukB* was 0.93 ± 0.16 (Mean ± SEM) (Figure 3B). The MukB mutation conferred a significant proliferation disadvantage in the systemic circulation in vivo. As shown in Figure 2B, the *mukB* mutation delays the time to death but does not affect the lethality of mice. Considering these results together, MukB of *V. vulnificus* is the necessary for rapid proliferation of *V. vulnificus* only in the systemic circulation.

### 3.3. The MukB Mutation Does Not Affect the Proliferation of V. vulnificus at the Local Infection Site

By analyzing the in vivo imaging system (IVIS) after subcutaneous inoculation of luciferase expression strains into the right caudal thighs of mice, bioluminescent signals were detected in a time course analysis. There was no difference in the bioluminescent signals from WT, *mukB*::Tn, and *mukB*::Tn/p*mukB* (Figure 4). These peaked at 6 h to 9 h post-inoculation, and gradually weakened to 12 h. However, the bioluminescent signals from the nonencapsulated strain E4 usually eliminated in vivo had disappeared entirely at 3 h post-inoculation (Figure 4A). Moreover, there were no significant differences in the bacterial burdens of muscle tissue localized beneath the *V. vulnificus* inoculation site at 6 h post-infection between WT, *mukB*::Tn, and *mukB*::Tn/p*mukB* (Figure 4B). Thus, the results demonstrated that the MukB mutation does not affect the proliferation of *V. vulnificus* at the local infection site.

## 4. Discussion

Although aggressive surgery and antibiotic therapy prolongs survival [18], *V. vulnificus* septicemia is considerably lethal because of the rapid dissemination of this bacterium in an infected person. Effective treatment for septicemia caused by this bacterium has been desired for some time. Toward this goal, we have tried to find factors involved in the rapid proliferation of *V. vulnificus* after its invasion into the systemic circulation.

The mean survival time of mice had increased for *mukB*::Tn infections than in other strains (Figure 2B). On the other hand, the *mukB* mutation did not affect the proliferation of *V. vulnificus* at the local infection site and mortality of mice after the subcutaneous inoculation (Figure 2B and Figure 4). Our data strongly suggested that there is a factor or factors related to the *mukB*::Tn mutation that can delay the proliferation of *V. vulnificus* in the systemic circulation but not at the local infection site. There are some possible candidates of these factors in *V. vulnificus* infected mouse. It was reported that MukB in *E. coli* play a central role in segregation of chromosome, and MukB null mutants failed to grow at over 30 °C [19]. It is also known that the core body temperature (systemic circulation) of the infected mouse should rise to around 40 °C due to the inflammation and decrease to around 30 °C with sepsis progresses. Considering with our data and these knowledges together, the *mukB*::Tn of *V. vulnificus* fail to grow at normal body temperature, but may grow at low body temperature in the late stage of sepsis (30 °C or below). Therefore, we compared the temperature sensitivity of *mukB*::Tn with those of WT and *mukB*::Tn/p*mukB* at 25, 37 and 41 °C several times. However, unfortunately, the data lack reproducibility. Thus, in the present stage, it has been not clear whether the rapid proliferation of *V. vulnificus* in vivo depending on MukB is affected by body temperature or not.

Another possibility is that the *mukB*::Tn can proliferate in the late stage of sepsis because of the suppression of a systemic circulation specific immune response, although *mukB*::Tn is eliminated by that immune response easier than WT before reaching the late stage. It has reported that immune suppression occurs in the late stage of sepsis [20]. Actually, we have reported that apoptosis of lymphocytes and macrophages was induced during *V. vulnificus* infection [21,22]. Lymphocyte depletion in peripheral blood by apoptosis was primarily associated with bacterial growth in vivo [22], suggesting that immune suppression in systemic circulation will occur. Future studies should address the kinds of immune responses that are involved.

This is the first report of the gene required for rapid proliferation of *V. vulnificus* after invasion into the systemic circulation. Our findings may provide clues into the development of new therapies for late stage of sepsis resulting from *V. vulnificus* infection.

## Figures and Tables

**Figure 1 microorganisms-09-00934-f001:**
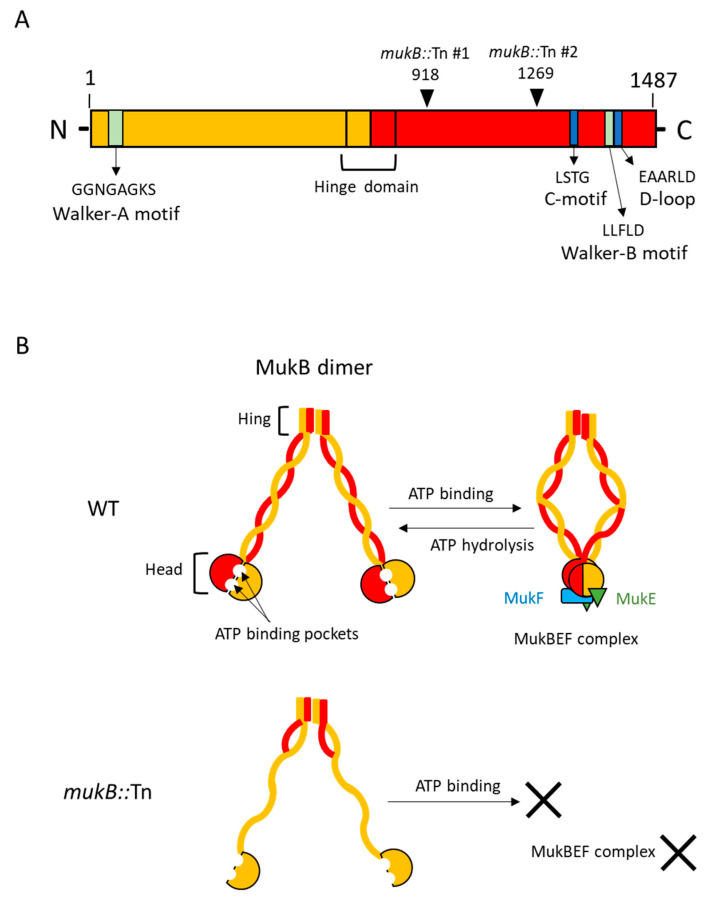
Prediction of the structure-function relationship of *mukB*::Tn. (**A**) Second structure of *V. vulnificus* MukB. Location and the amino acid sequences of Walker A, C motif, Walker B, D loop, and hinge domain were represented. Arrowheads indicate the predicted expression site of each *mukB*::Tn clone. (**B**) Structure and function relationship of MukB expressed in WT and *mukB*::Tn #1.

**Figure 2 microorganisms-09-00934-f002:**
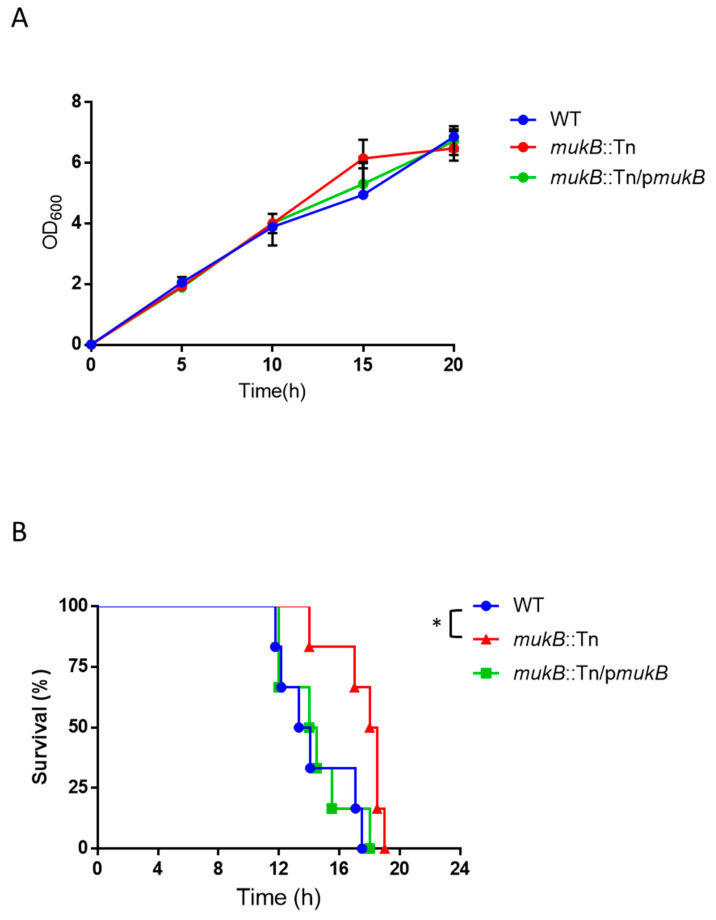
Growth of *V. vulnificus* and survival of mice. (**A**) In vitro growth of *V. vulnificus* WT, *mukB*::Tn, and *mukB*::Tn/p*mukB*. The starting optical density at 600 nm (OD_600_) of the cultures was adjusted around 0.01. OD_600_ was measured every 5 h. (**B**) Survival curve analysis. Kaplan-Meier survival curves for mice inoculated subcutaneously with WT (*n* = 6), *mukB*::Tn (*n* = 6), or *mukB*::Tn/p*mukB* (*n* = 6) and monitored for 24 h. Statistical analyses: log-rank test (* *p* < 0.05).

**Figure 3 microorganisms-09-00934-f003:**
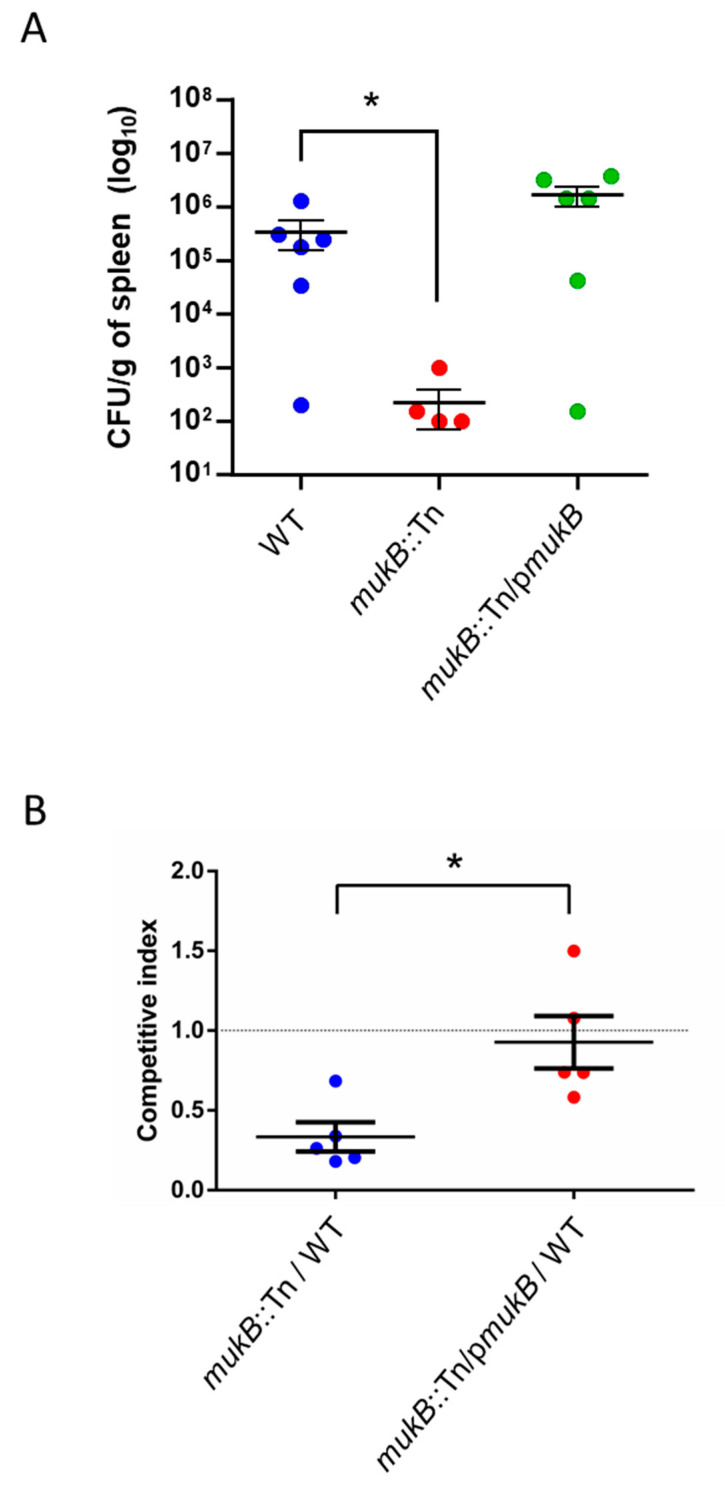
MukB is the necessary gene for proliferation of *V. vulnificus* in the systemic circulation. (**A**) Bacterial burdens in the spleen of mice subcutaneously inoculated with WT, *mukB*::Tn, or *mukB*::Tn/p*mukB* assessed as CFU/g at 12 h post-infection. Each symbol represents an individual mouse (*n* = 6/group). Error bars indicate standard error of mean (SEM). Statistical analyses: Kruskal-Wallis with Dunn’s multiple comparison (* *p* < 0.05). (**B**) Bacterial burdens in the spleen of mice subcutaneously inoculated with WT, *mukB*::Tn, and *mukB*::Tn/p*mukB* assessed as CFU/g at 6 h post-infection. Each symbol represents an individual mouse (*n* = 5/group). Error bars indicate SEM. Statistical analyses: Mann-Whitney *U*-test (* *p* < 0.05). CFU, Colony forming unit.

**Figure 4 microorganisms-09-00934-f004:**
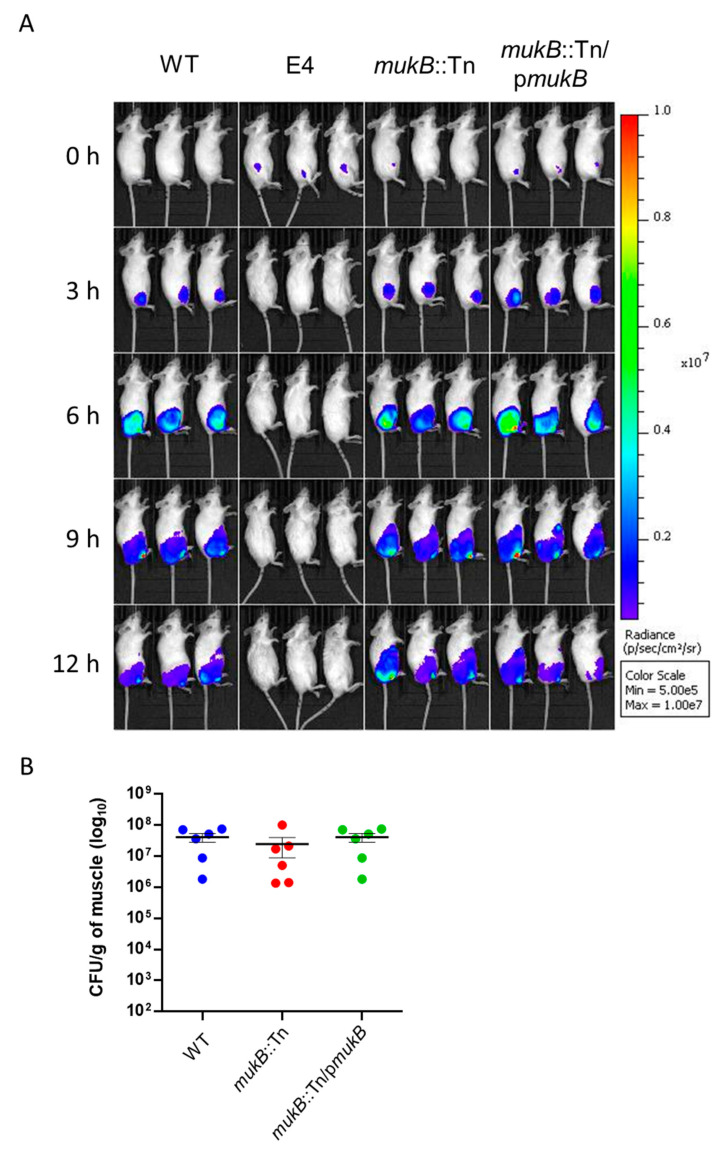
MukB mutation does not affect the proliferation of *V. vulnificus* at the local infection site. (**A**) Bacterial spread through the subcutaneous tissue. The luminescence signal of WT, E4 (nonencapsulated strain), *mukB*::Tn, and *mukB*::Tn/p*mukB* was detected every 3 h by IVIS during the 12-h time course. (**B**) Burden of WT, *mukB*::Tn, and *mukB*::Tn/p*mukB* infections in the muscle beneath the subcutaneous inoculation site. The degree of burden was assessed as CFU/g at 12 h post-infection. Each symbol represents an individual mouse (*n* = 6/group). Error bars indicate standard error of mean (SEM). No significant difference among bacterial burdens in the three groups. CFU, Colony forming unit.

## Data Availability

The data that support the findings of this study are available from the corresponding author, [author initials], upon reasonable request.

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
