# Peer review of "MukB Is a Gene Necessary for Rapid Proliferation of Vibrio vulnificus in the Systemic Circulation but Not at the Local Infection Site in the Mouse Wound Infection Model"

_microorganisms, 2021, doi:10.3390/microorganisms9050934_

Round 1

Reviewer 1 Report

Authors mentions they report some "factor/s" but have not really identified any specific factor/s related to mukB::Tn. It seems kind of vague, authors need to change the language of abstract and main text. 

Secondly, they should attempt to identify the potential factors which they claim is involved in mukB:tn strain. In present for study seems vey superficial.

Author Response

To the Reviewer 1  

Thank you for your careful reading and for giving us the kind suggestions. We have tried and keep trying to identify the factor, which is involved in mukB::Tn strain. Unfortunately, we could not find it so far. We have taken the comments and suggestions for your comments into account in the revised manuscript below.

Comment

Authors mentions they report some "factor/s" but have not really identified any specific factor/s related to mukB::Tn. It seems kind of vague, authors need to change the language of abstract and main text. 

Answer: Thank you for your suggestion. We changed the word from factor (s) to gene in statements related to MukB function in the revised manuscript including the manuscript title.

Secondly, they should attempt to identify the potential factors which they claim is involved in mukB:tn strain. In present for study seems vey superficial.

Answer: Thank you for your comment. We have tried and trying to identify the factor, which is involved in mukB::Tn strain. Unfortunately, we could not find it so far.

Reviewer 2 Report

In this manuscript titled “MukB is a factor necessary for rapid proliferation of Vibrio vulnificus in the systemic circulation but not at the local infection site in the mouse wound infection model’ Kashimoto et al establish a potential causal effect of mukB in rapid proliferation of V. vulnificus in systemic infection/circulation and not involved in proliferation at the site of infection. They have taken a genetic approach in identifying a transposon insertion in mukB leading to increased time to death in a murine model and is reversed by in trans complementation of the mukB on a plasmid. They also establish that the mutant is not affected in in vitro growth properties. Although the mutant affects systemic proliferation and increased time to death, it does not affect lethality; i.e., prevent death. This study suggests that mukB may a potential target for novel drugs against V. vulnificus infection.

It is a very well-conceived study and the manuscript is well written and concise. The Figures and descriptions are clearly conveyed. Overall, this reviewer has no major comments except, the link between the genotype to phenotype is not very well explained in the discussion. That is, what is the mechanism of how inactivation of mukBleads to the observed effect in systemic proliferation? The authors have attempted to explain that this may be attributed to temperature fluctuations during septicemia and lack of expression of wild type mukB at low temperatures seen in E.coli may be mimicked by mukB mutation.  Again, this is not very clear and perhaps requires further experiments to show directly in V. vulnificus.

I have a few minor suggestions:

Line 19-20: we found a lower bacterial burden of mukB::Tn in the spleen than that ‘in the’ WT strain infecting the systemic- remove in the and replace with ‘of’

Line 33: replace ‘invasion’ with ‘entry’

Line 46: replace ‘close’ with ‘closed’

Line 49: replace is the ‘necessary factor’ with ‘is one of the necessary factors’

Line 63-65: “When required, the medium was supplemented with rifampicin (63)

(50 μg/mL), chloramphenicol (10 μg/mL), or ampicillin (100 μg/mL) for V. vulnificus to 64

maintain the pXen-13 plasmid for In vivo imaging system and the pACYC plasmid for 65

mukB complementation.- question: Which antibiotic for what? Rifampicin for strain, Chloramphenicol for pXen-13 and ampicillin for pACYC ? please specify?

Line 86: 10uM- is this correct unit? Or is it 10ug?

Line 150: ‘data were tested’ replace with ‘data were analyzed’

Line 165: WT, mukB::Tn and pmukB: the pmukB- do you mean to say mukB:Tn/pmukB; it can’t be a plasmid by itself and it has to be a strain containing the plasmid. This is repeated in many other places- like Line 192.and in Figure 2 and other places.

Line 207: two mutant clones of MukB by inserting a transposon- two mutant clone of MukB that contained a transposon

Lines 216-217: Each mukB::Tn mutant, which was obtained by our STM assay, predicted the ex- 216

pression up to 918th lysine

DO you mean to say: Each mukB::Tn mutant, which was obtained by our STM assay, resulted in truncated protein from residues1-918 lysine (mukB::Tn #1) and 1-1269th glycine (mukB::Tn #2),

Line 221: The MukBEF complex could not form in both- change to ‘The MukBEF complex is not expected to form in both

Line 224: the mukB complement strain; it has to be written out (mukB::Tn#1/pmukB)

Line 259: mukB mutation leads to delay of lethal time of mice, but not affected the lethality; change to mukB mutation delays the time to death but does not affect the lethality

Figure 4B; Why the E4 mutant not included in the figure? What is the CFU in that mutant? 0 are not counted or what?

Author Response

To the Reviewer 2   

Thank you for your suggestions. We have taken the comments and suggestions for your comments into account in the revised manuscript below.

Comments and Suggestions for Authors

In this manuscript titled “MukB is a factor necessary for rapid proliferation of Vibrio vulnificus in the systemic circulation but not at the local infection site in the mouse wound infection model’ Kashimoto et al establish a potential causal effect of mukB in rapid proliferation of V. vulnificus in systemic infection/circulation and not involved in proliferation at the site of infection. They have taken a genetic approach in identifying a transposon insertion in mukB leading to increased time to death in a murine model and is reversed by in trans complementation of the mukB on a plasmid. They also establish that the mutant is not affected in in vitro growth properties. Although the mutant affects systemic proliferation and increased time to death, it does not affect lethality; i.e., prevent death. This study suggests that mukB may a potential target for novel drugs against V. vulnificus infection.

It is a very well-conceived study and the manuscript is well written and concise. The Figures and descriptions are clearly conveyed. Overall, this reviewer has no major comments except, the link between the genotype to phenotype is not very well explained in the discussion. That is, what is the mechanism of how inactivation of mukB leads to the observed effect in systemic proliferation? The authors have attempted to explain that this may be attributed to temperature fluctuations during septicemia and lack of expression of wild type mukB at low temperatures seen in E.coli may be mimicked by mukB mutation.  Again, this is not very clear and perhaps requires further experiments to show directly in V. vulnificus.

Answer: Thank you for your careful reading and for giving us the kind suggestions. We have tried and keep trying to identify the factor, which is involved in mukB:Tn strain. Unfortunately, we could not find it so far. We have taken the comments and suggestions for your comments into account in the revised manuscript below.

I have a few minor suggestions:

Line 19-20: we found a lower bacterial burden of mukB::Tn in the spleen than that ‘in the’ WT strain infecting the systemic- remove in the and replace with ‘of’

Answer: According to the reviewer's suggestion, we changed the statement on p2, L21-23 in the revised manuscript as follows. "we found a lower bacterial burden of mukB::Tn in the spleen than that of the WT strain infecting the systemic circulation."

Line 33: replace ‘invasion’ with ‘entry’

Answer: Thank you for your suggestion. In our previous study, we found that V. vulnificus invades deep soft tissue and systemic circulation by their chemotactic ability (Virulence. 2020. 11:840-848.). So, we would like to write "invasion" in the revised manuscript.

Line 46: replace ‘close’ with ‘closed’

Answer: Thank you for your comment. "Close" have been replaced with "closed" on p3, L56 in the revised manuscript.

Line 49: replace is the ‘necessary factor’ with ‘is one of the necessary factors’

Answer: Thank you for your comment. " necessary factor " have been replaced with " is one of the necessary genes" on p3, L59, p10, L243 and L247 in the revised manuscript.

Line 63-65: “When required, the medium was supplemented with rifampicin (63)

(50 μg/mL), chloramphenicol (10 μg/mL), or ampicillin (100 μg/mL) for V. vulnificus to 64

maintain the pXen-13 plasmid for In vivo imaging system and the pACYC plasmid for 65

mukB complementation.- question: Which antibiotic for what? Rifampicin for strain, Chloramphenicol for pXen-13 and ampicillin for pACYC ? please specify?

Answer: We changed the statement about the role for each antibiotics more clearly on p4, L73-76 in the revised manuscript as follows. "When required, the medium was supplemented with rifampicin (50 µg/mL) to selective grow for V. vulnificus, chloramphenicol (10 µg/mL) for maintain the pACYC, or ampicillin (100 µg/mL) for maintain the pXen-13 plasmid in In vivo imaging system. The pACYC plasmid was used for mukB complementation."

Line 86: 10uM- is this correct unit? Or is it 10ug?

Answer: 10uM is correctly.

Line 150: ‘data were tested’ replace with ‘data were analyzed’

Answer: Thank you for your suggestion. The"data were tested" was replaced with "Data were analyzed" on p.7, L166 in the revised manuscript.

Line 165: WT, mukB::Tn and pmukB: the pmukB- do you mean to say mukB:Tn/pmukB; it can’t be a plasmid by itself and it has to be a strain containing the plasmid. This is repeated in many other places- like Line 192.and in Figure 2 and other places.

Answer: Yes. The pmukB means the mukB::Tn/pmukB. We changed from "the pmukB" to "the mukB::Tn/pmukB" throughout the entire of revised manuscript.

Line 207: two mutant clones of MukB by inserting a transposon- two mutant clone of MukB that contained a transposon.

Answer: The statement was changed as you pointed out on p9, L222- p10, 223 in the revised manuscript.

Lines 216-217: Each mukB::Tn mutant, which was obtained by our STM assay, predicted the expression up to 918th lysine. Do you mean to say: Each mukB::Tn mutant, which was obtained by our STM assay, resulted in truncated protein from residues1-918 lysine (mukB::Tn #1) and 1-1269th glycine (mukB::Tn #2),

Answer: Thank you for your kind suggestion. The statement was changed as you pointed out on p10, L232-233 in the revised manuscript.

Line 221: The MukBEF complex could not form in both- change to ‘The MukBEF complex is not expected to form in both-

Answer: Thank you. The statement was changed as you pointed out on p10, L236 in the revised manuscript.

Line 224: the mukB complement strain; it has to be written out (mukB::Tn#1/pmukB)

Answer: Thank you for your kind suggestion. The statement was changed as mukB::Tn /pmukB for complement strain on p10, L240 in the revised manuscript.

Line 259: mukB mutation leads to delay of lethal time of mice, but not affected the lethality; change to mukB mutation delays the time to death but does not affect the lethality.

Answer: The statement was changed as you pointed out on p11, L263 in the revised manuscript.

Figure 4B; Why the E4 mutant not included in the figure? What is the CFU in that mutant? 0 are not counted or what?

Answer: The E4 strain was used as the negative control in IVIS assay, which is disappeared within a short time. So, we have not counted the number of E4 strain in the muscle tissue.

Reviewer 3 Report

MukB is a factor necessary for rapid proliferation of Vibrio vulnificus in the systemic circulation but not at the local infection site in the mouse wound infection model

Dear Authors,

This manuscript presents a factor, MukB, required for the rapid proliferation of V. vulnificus in the systemic circulation and potentially required for its survival, by using a murine infection model. These findings may provide clues into the development of new therapies for late stage sepsis resulting from V. vulnificus human infection.

The manuscript is well written and organised, comprising the abstract, introduction, material and methods with subheadings, results with subheadings, discussion and references.

Please see below some suggestions for improvement.

  1. Materials and methods

Lines 77, 142 and 147 – Please harmonize abbreviation for “optical density at 600nm” by using OD600.

Lines 79, 157, 186 - Please include abbreviation for “subcutaneously” (s.c.) before use.

2.3. Generation of transposon mutants and identification of the Tn insertion gene

Line 80: in order to make the text more comprehensible to readers working outside this topic, please replace “CFU” by “colony forming unit (CFU)”, as this abbreviation is not explained in the text (only explained in Figs. 3 and 4).

2.5 Mice

Please, describe mice housing conditions during breeding and maintenance, such as light cycle, feeding and water sources. Also, please clarify the supplier of the mice strains in the sentences below.

Lines 135-136: state “mice were purchased from Charles River Laboratories Japan (Atsugi, Japan).”But, Lines 149-150: refer “mice (5 weeks, C57BL/6, female, CLEA, Japan).”

  1. Results

Line 135: please replace “SEM” by “standard error of mean (SEM)”, as this abbreviation is not explained in the text nor in Figs. 3 and 4.

  1. Discussion

Lines 311-313: “the number of colonies of mukB::Tn was significantly decreased to about half compared with those of both WT and pmukB at all tested temperatures (data not shown)”. Please, provide these data as Supplementary Material.

Reference

The list of references is valuable and the correspondence with the citations is good, except for the following two, which are not cited in the manuscript:

Lines 362-364: “16. Woo, J.S.; Lim, J.H.; Shin, H.C.; Suh, M.K.; Ku, B.; Lee, K.H.; Joo, K.; Robinson, H.; Lee, J.; Park, S.Y.; Ha, N.C.; Oh, B.H. Structural studies of a bacterial condensin complex reveal ATP-dependent disruption of intersubunit interactions. Cell. 2009, 136, 85-96. doi: 10.1016/j.cell.2008.10.050.”

Lines 365-366: “17. Zawadzka, K.; Zawadzki, P.; Baker, R.; Rajasekar, K.V.; Wagner, F.; Sherratt, D.J.; Arciszewska, L.K. MukB ATPases are regulated independently by the N- and C-terminal domains of MukF kleisin. eLife, 2018, 7, doi: 10.7554/eLife.31522.

Best Regards, Reviewer

Author Response

To the Reviewer 3  

Thank you for your careful reading and for giving us the kind suggestions. We have taken the comments and suggestions for your comments into account in the revised manuscript below.

Comments and Suggestions for Authors

This manuscript presents a factor, MukB, required for the rapid proliferation of V. vulnificus in the systemic circulation and potentially required for its survival, by using a murine infection model. These findings may provide clues into the development of new therapies for late stage sepsis resulting from V. vulnificus human infection.

The manuscript is well written and organised, comprising the abstract, introduction, material and methods with subheadings, results with subheadings, discussion and references.

Please see below some suggestions for improvement.

  1. Materials and methods

Lines 77, 142 and 147 – Please harmonize abbreviation for “optical density at 600nm” by using OD600.

Answer: Thank you for your pointing it out. We have harmonized these by using OD600.

Lines 79, 157, 186 - Please include abbreviation for “subcutaneously” (s.c.) before use.

Answer: We corrected this abbreviation as you pointed out on p7, L173- 174 in the revised manuscript.

2.3. Generation of transposon mutants and identification of the Tn insertion gene

Line 80: in order to make the text more comprehensible to readers working outside this topic, please replace “CFU” by “colony forming unit (CFU)”, as this abbreviation is not explained in the text (only explained in Figs. 3 and 4).

Answer: We corrected this abbreviation as you pointed out on p4, L89- p5, L90 in the revised manuscript.

   2.5 Mice

Please, describe mice housing conditions during breeding and maintenance, such as light cycle, feeding and water sources. Also, please clarify the supplier of the mice strains in the sentences below.

Lines 135-136: state “mice were purchased from Charles River Laboratories Japan (Atsugi, Japan).” But, Lines 149-150: refer “mice (5 weeks, C57BL/6, female, CLEA, Japan).”

Answer: Thank you for your pointing it out. Charles River Laboratories is correctly. In addition, we added the statement of mice housing conditions on p7, L147-150 in the revised manuscript as follows." The mice were housed in plastic cages in a group and were maintained on a standard laboratory diet (rat chow MF, Oriental Yeast Co., Ltd. Tokyo, Japan) and tap water under a 12 h light and dark cycle. Ambient temperature during the study was maintained at about 21°C."

  1. Results

Line 135: please replace “SEM” by “standard error of mean (SEM)”, as this abbreviation is not explained in the text nor in Figs. 3 and 4.

Answer: We corrected this abbreviation as you pointed out on p11, L260 in the revised manuscript.

Discussion

Lines 311-313: “the number of colonies of mukB::Tn was significantly decreased to about half compared with those of both WT and pmukB at all tested temperatures (data not shown)”. Please, provide these data as Supplementary Material.

Answer: Thank you for your suggestion. We confirmed the data about the temperature sensitivity between WT, mukB::Tn and mukB::Tn/pmukB several times. Unfortunately, however, we could not get the data with good reproducibility. So, we described this on p.17, L331-333 in the revised manuscript as follows. "However, unfortunately, the data lack reproducibility. Thus, in the present stage, it has been not clear whether the rapid proliferation of V. vulnificus in vivo depending on MukB is affected by body temperature or not."

Reference

The list of references is valuable and the correspondence with the citations is good, except for the following two, which are not cited in the manuscript:

Lines 362-364: “16. Woo, J.S.; Lim, J.H.; Shin, H.C.; Suh, M.K.; Ku, B.; Lee, K.H.; Joo, K.; Robinson, H.; Lee, J.; Park, S.Y.; Ha, N.C.; Oh, B.H. Structural studies of a bacterial condensin complex reveal ATP-dependent disruption of intersubunit interactions. Cell. 2009, 136, 85-96. doi: 10.1016/j.cell.2008.10.050.”

Lines 365-366: “17. Zawadzka, K.; Zawadzki, P.; Baker, R.; Rajasekar, K.V.; Wagner, F.; Sherratt, D.J.; Arciszewska, L.K. MukB ATPases are regulated independently by the N- and C-terminal domains of MukF kleisin. eLife, 2018, 7, doi: 10.7554/eLife.31522.

Answer: Both references were cited on p10, L225 in the revised manuscript.

Reviewer 4 Report

1.  lines 65 & 272: "In vivo" --> in vivo

2. line 75: "Kanamycin" --> kanamycin

3. line 77: "optical density at 600 nm" is defined as OD600

lines 141-142 it is again defined as OD600

line 147: OD 600

line 168 & 193: OD600

line 226: optical density at 600nm

Be consistant. Define the abbreviation at line 77, and use that abbreviation thereafter.

4. line 137: "for other all" --> for all other

5. line 160: "800 rpm" Convert this to x g.

6. Discussion: Remove references to figures and materials and methods. Move the material on growth at different temperatures to the results section.

7.  line 330:  Who is the founder?

Author Response

To the Reviewer 4 

Thank you for your careful reading and for giving us the kind suggestions. We have taken the comments and suggestions for your comments into account in the revised manuscript below.

Comments and Suggestions for Authors

  1. lines 65 & 272: "In vivo" --> in vivo

Answer: We have corrected these errors. Thank you.

  1. line 75: "Kanamycin" --> kanamycin

Answer: We have corrected this error. Thank you.

  1. line 77: "optical density at 600 nm" is defined as OD600

lines 141-142 it is again defined as OD600

line 147: OD 600

line 168 & 193: OD600

line 226: optical density at 600nm

Be consistant. Define the abbreviation at line 77, and use that abbreviation thereafter.

Answer: We have harmonized this error as OD600. Thank you.

  1. line 137: "for other all" --> for all other

Answer: Thank you for your pointing it out. We have corrected this on p7, L146 in the revised manuscript.

  1. line 160: "800 rpm" Convert this to x g.

Answer: We corrected 800 rpm to 42 x g.

  1. Discussion: Remove references to figures and materials and methods. Move the material on growth at different temperatures to the results section.

Answer: Thank you for your suggestion. We confirmed the data about the temperature sensitivity between WT, mukB::Tn and mukB::Tn/pmukB several times. Unfortunately, however, we could not get the data with good reproducibility. So, we described this on p.17, L331-333 in the revised manuscript as follows. "However, unfortunately, the data lack reproducibility. Thus, in the present stage, it has been not clear whether the rapid proliferation of V. vulnificus in vivo depending on MukB is affected by body temperature or not."

  1. line 330:  Who is the founder?

Answer: We have corrected this error.